# MAKING THE SUBSPACE ASSUMPTION WORK: LOW-DIMENSIONAL EXPLORATION FOR HIGH-DIMENSIONAL BAYESIAN OPTIMIZATION

## ABSTRACT

Bayesian optimization (BO) has been widely used from algorithm hyperparameter tuning to emerging scientific applications. However, its performance degradation in high-dimensional settings remains a long-standing bottleneck. Recent studies suggest that standard BO can remain competitive in high dimensions by carefully tuning priors or initialization, which has shifted attention away from subspace-based methods. We argue that the limitations of existing subspace methods stem not from the subspace assumption itself, but from the lack of an effective balance between thoroughly exploiting the current subspace and expanding to larger ones. To address this, we propose a high-dimensional Bayesian optimization algorithm, which projects the input-space into a lower dimensional subspace and consequently expands the subspace dimension based on cumulative regret minimization. Our method allocates evaluation budgets linearly according to the subspace dimension, thereby fully exploiting structural information before expansion. Our experimental evaluations show that our method significantly outperforms existing state-of-the-art baselines on several challenging high-dimensional synthetic and real-world tasks, highlighting the continued potential of subspace methods in high-dimensional Bayesian optimization.

## 1 INTRODUCTION

Bayesian Optimization (BO) has emerged as a key enabler of AI for Science due to its sample efficiency in optimizing expensive black-box functions. In scientific domains such as materials discovery (Lookman et al., 2019; Farache et al., 2022; Li et al., 2023), drug design (Yang et al., 2025; Clarke et al., 2008), and polygenic risk prediction (Ge et al., 2019; Li et al., 2022; Privé et al., 2020), practitioners often introduce a large number of potentially relevant variables to enhance predictive and optimization performance. Similarly, complex machine learning systems—for example, jointly tuning the kernel and regularization strength of a support vector machine (Snoek et al., 2012), or co-optimizing the policy network architecture (Lizotte, 2008) and control parameters (Brochu et al., 2010) in reinforcement learning—routinely involve high-dimensional parameter spaces. This has led to the phenomenon of artificially constructed high dimensionality, where the number of variables can reach thousands or even tens of thousands. The ubiquity of such settings poses serious challenges to the efficiency and scalability of traditional BO methods, underscoring the urgent need for advanced high-dimensional optimization techniques.

In these ultra-high-dimensional regimes, the number of evaluations required to adequately explore the search space grows exponentially with dimensionality, making Gaussian process (GP) surrogates ineffective at modeling the objective. This leads acquisition functions to lose guidance, a manifestation of the curse of dimensionality. One class of approaches mitigates this issue by introducing structural assumptions to reduce problem complexity. Among them, subspace methods posit that the function depends primarily on a low-dimensional embedding of the input space (Wang et al., 2016; Letham et al., 2020; Nayebi et al., 2019), while BAxUS (Papenmeier et al., 2022b) incrementally expands the subspace dimensionality. Another line of work, additive methods, assumes the objective decomposes into a sum of low-dimensional components, enabling localized optimization.

Recent work, however, has questioned the necessity of such assumptions. It suggests that poor performance in high dimensions often arises from inappropriate lengthscale priors or initialization,

which make the GP surrogate overly complex and bias the acquisition function toward local optima. By modifying kernel lengthscale priors (Eriksson & Jankowiak, 2021; Hvarfner et al., 2024) or improving initialization strategies (Xu et al., 2024), some standard BO variants have demonstrated surprisingly strong performance on high-dimensional benchmarks—sometimes rivaling or surpassing structure-exploiting methods. These findings have sparked debate over whether structural assumptions are truly essential. Nevertheless, the recent unstructured BO methods that report strong performance in high dimensions (Eriksson & Jankowiak, 2021; Hvarfner et al., 2024) still face fundamental scalability challenges. Both rely on Sobol initialization, which is subject to a strict dimensionality cap and becomes unavailable in ultra-high-dimensional regimes, and both adopt ARD kernels, which exacerbate scalability issues by introducing instability, excessive computational costs, and degraded acquisition performance.

To address this, we propose **GRABBO** (Guided Regret-Aware Bayesian Subspace Optimization), a framework that integrates theoretical guidance with efficient subspace-based exploration. The key insight is that subspace expansion should be driven by exploration quality rather than fixed schedules. We introduce a simple yet effective regret-based budget allocation rule, theoretically motivated by the relationship between subspace dimensionality and cumulative regret, which encourages sufficient exploration of each subspace before expansion and improves both stability and efficiency.

Our main contributions are:

1. We show that one of the state-of-the-art subspace methods, BAxUS, expands dimensionality prematurely, failing to fully exploit lower-dimensional subspaces and thereby leading to inefficient and unstable optimization.
2. To address this, we propose a budget-driven expansion strategy with a regret-based linear rule, theoretically motivated to allocate the evaluation budget in proportion to the subspace dimensionality and ensure thorough exploration.
3. We present GRABBO, a principled framework for nested subspace optimization that operationalizes our budget-aware strategy.
4. Through comprehensive evaluations, we demonstrate that GRABBO substantially outperforms state-of-the-art methods. We highlight two key advantages: superior efficiency in the low-budget regime by rapidly finding high-quality solutions, and unique robustness in ultra-high dimensions where competing methods stagnate.

## 2 RELATED WORK

**Bayesian Optimization.** Bayesian optimization (BO) is a sequential and sample-efficient framework for optimizing expensive black-box functions, where a surrogate model is iteratively fit, queried, and updated to guide the search (Mockus et al., 1978; Brochu et al., 2010). At each iteration, it fits a probabilistic surrogate model (typically a Gaussian Process) to the previously collected observations, then selects the next evaluation point by optimizing an acquisition function that trades off between exploration and exploitation. The newly observed data is added to the dataset, and the surrogate is updated, and this process repeats until the evaluation budget is exhausted. While BO is highly effective in low dimensional settings, its performance deteriorates in high dimensions due to challenges in kernel estimation, posterior inference, and acquisition maximization. This has spurred a range of methods to improve the scalability of BO by introducing structural assumptions, adapting the kernel, or localizing the search.

**Structure-Assuming Methods.** Structure-assuming approaches tackle high-dimensional BO by reducing the problem's intrinsic dimensionality through structural assumptions. Additive models decompose the objective into a sum of low-dimensional components (Kandasamy et al., 2015), typically by partitioning the input variables into fixed or adaptive groups. Some implementations further allow overlapping groups via dependency graph structures (Rolland et al., 2018), enabling more expressive modeling.

Another commonly explored direction assumes that the objective function varies meaningfully only along a low-dimensional subspace embedded in the input space. REMBO (Wang et al., 2016) and HeSBO (Nayebi et al., 2019) implement this idea using random projections to embed the original space into a lower-dimensional one, where BO is performed. ALEBO (Letham et al., 2020) improves projection quality by learning a Mahalanobis metric and enforcing linear constraints during acquisition optimization, ensuring that proposed candidates remain within the feasible domain.

Extending this idea further, dynamic subspace methods such as BAxUS (Papenmeier et al., 2022b), Bounce (Papenmeier et al., 2023), and BOIDS (Ngo et al., 2025), adapt the embedding dimension over time based on observed optimization progress. Starting from a small subspace, these methods gradually expand the search space only when optimization progress stalls, maintaining efficiency while preserving the ability to recover the global optimum.

**Lengthscale-adaptive methods.** Instead of committing to an assumed structure, these approaches place informative priors or initializations on the GP lengthscales and update them over time based on observed data. SAASBO (Eriksson & Jankowiak, 2021) infers sparsity in the GP prior, effectively pruning irrelevant dimensions. VBO (Hvarfner et al., 2024) imposes LogNormal priors over lengthscales to improve stability, while SBO (Xu et al., 2024) initializes the lengthscales as a function of the input dimension to prevent them from becoming too small during optimization; both outperform methods that assume a fixed structure.

**Trust-region-based methods.** To improve the reliability of surrogate models in high-dimensional spaces, several approaches introduce trust regions to localize the search and stabilize model behavior. TuRBO (Eriksson et al., 2019) maintains multiple parallel trust regions that adaptively expand or contract based on observed performance. This localized search paradigm reduces the reliance on global surrogate accuracy and improves robustness. BAxUS and BOIDS also incorporate trust-region mechanisms as part of their dynamic subspace frameworks.

**BAxUS in focus.** Among all the subspace-based methods, BAxUS (Papenmeier et al., 2022b) serves as a representative baseline. It begins optimization in a low-dimensional embedding formed by randomly partitioning the input features, and maintains a local trust region to ensure reliable surrogate behavior. When no improvement is observed over consecutive iterations, BAxUS expands the embedding dimension, eventually covering the full space if necessary. This combination of progressive subspace expansion and localized search enables efficient exploration without sacrificing global optimality. We analyze the design of BAxUS in Section 3 and revisit its behavior in Section 3.2, which together motivate our proposed improvements.

## 3 ADDITIONAL BACKGROUND ON BAxUS

In this section, we provide additional background on the BAxUS method that is relevant to understanding our approach, as well as the key limitations of BAxUS that our method aims to address.

### 3.1 DETAILS OF BAxUS

In this section, we provide context for BAxUS by revisiting HeSBO and its sparse projection. We then outline the two design choices in BAxUS: balanced partitioning of features and expansion of the subspace triggered by trust region shrinkage. Both HeSBO and BAxUS employ a highly sparse projection matrix $S \in \{0, \pm 1\}^{d \times D}$ that maps each original coordinate to exactly one subspace coordinate, which is equivalent to each column of $S$ having a single non-zero entry. A subspace point $y \in \mathbb{R}^d$ is then mapped back to the original space via $x = S^\top y$. The choice of the $\{0, \pm 1\}$ alphabet is deliberate, as it ensures that after standard box normalization, the back-projected point $x$ remains within the original domain componentwise. Furthermore, under the random $\pm 1$ signs and uniform assignment, pairwise distances are preserved in expectation. The two methods differ critically in how this assignment of original coordinates to subspace coordinates is determined.

In HeSBO, the assignment is performed randomly and independently for each of the $D$ original coordinates, mapping each one to a dimension selected uniformly at random from the $d$ available subspace dimensions. While simple, this approach can induce a finite-sample imbalance, where some subspace coordinates aggregate a disproportionate number of original coordinates while others aggregate very few, creating potential information bottlenecks. Moreover, the subspace dimension $d$ must be fixed a priori, creating a difficult trade-off where a dimension that is too small may cause the subspace optimum to deviate from the true optimum, while one that is too large diminishes the benefits of the subspace assumption.

BAxUS addresses these limitations through two primary innovations. First, to mitigate the imbalance, it constructs the projection $S$ using a structured partitioning scheme. The $D$ original coordinates are divided into $d_i$ balanced groups, and all members of a group are aggregated with random $\pm 1$ signs to form the corresponding subspace coordinate. This ensures a more uniform distribution of original coordinates across the embedding. Second, to resolve the static trade-off on $d$, BAxUS

employs a nested-subspace framework governed by a trust region (TR). In this framework, the initial subspace dimension $d_1$ is determined by a hyperparameter, the expansion factor $b > 1$, in conjunction with the full dimension $D$. After $\tau_{\text{fail}}$ consecutive non-improving evaluations, the TR side length $L$ is shrunk. Once $L$ falls below a minimum threshold $L_{\min}$, the algorithm expands the subspace from $d_i$ to a new dimension $d_{i+1}$ according to the rule $d_{i+1} = \min(d_i \cdot b, D)$. This process involves redistributing the features that form the $d_i$ coordinates into a larger set of $d_{i+1}$ coordinates. Crucially, this expansion is lossless. For any previously evaluated point $y \in \mathbb{R}^{d_i}$, a corresponding point $y' \in \mathbb{R}^{d_{i+1}}$ can be constructed such that their projection to the original space is identical ($S'^\top y' = S^\top y$). This property allows BAxUS to increase dimensionality only when warranted by the search process, without discarding valuable evaluation data.

## 3.2 LIMITATIONS OF BAxUS

BAxUS incorporates many intuitive design choices that may introduce practical performance issues. Specifically, the mechanism that reduces the TR side length only after $\tau_{fail}$ consecutive failures is problematic. This tolerance is defined as $\tau_{fail} = \max\{1, \min(f_t, f_{t,\max})\}$, where the upper bound is $f_{t,\max} = \max\{4, d_i\}$ for a subspace of dimension $d_i$. The primary term $f_t$ is calculated as $f_t = \left\lceil \frac{\text{budget}}{2\gamma} \right\rceil$, where budget is the evaluation quota proportionally allocated to the current subspace, and $\gamma$ is a constant representing the number of halvings the TR undergoes to shrink from its initial to its minimum size. The underlying rationale is that the total number of evaluations required to trigger $k$ trust-region shrinkages—accounting for the fact that any successful step resets the failure counter—will probabilistically align with the allocated budget. The factor of 2 in the denominator likely serves as an empirical correction to compensate for this stochastic process.

However, this intuition-driven method is inherently brittle, with its behavior strongly affected by the unknown complexity of the objective function. Under the setting of a 1000-dimensional problem with a 1000-evaluation budget, expansion block size $b = 4$, and initial subspace dimension $d_1 = 1$, the mechanism exhibits two failure modes. The computed $\tau_{fail}$ values are 1 for dimensions between 1 and 16, 4 for 64 dimensions, 15 for 256 dimensions, and 57 for the full 1000 dimensions. For complex objectives where improvements are difficult to obtain, $\tau_{fail}$ is very small in the early, low-dimensional stages due to the small allocated budget. This makes it easy to trigger expansion—often rapidly jumping to the full space—after only a fewfailures. Once the full $D$-dimensional space is reached, however, $\tau_{fail}$ becomes very large, and achieving such a long sequence of failures is improbable. Consequently, the TR rarely shrinks, forcing the algorithm to search in a vast, high-dimensional, and complex region where local refinement is difficult, increasing the risk of convergence to suboptimal solutions.

For simple objectives, improvements are relatively easy to find, so the small $\tau_{fail}$ values in low-dimensional subspaces allow for more thorough exploration before expansion. Yet, once the full space is reached, the same problem arises: the large $\tau_{fail}$ suppresses TR shrinkage, trapping the search in an overly broad high-dimensional region and again undermining fine-grained optimization. This recurring inefficiency indicates that driving subspace expansion with a $\tau_{\text{fail}}$ based TR shrinkage trigger is likely suboptimal, and alternative mechanisms merit consideration.

## 4 THE GRABBO ALGORITHM

In this section, we introduce GRABBO and explain how it addresses two limitations of BAxUS identified in Section 3.2. BAxUS tends to expand subspace dimensionality too early during optimization, which limits exploration in low dimensions; it also fails to adapt in high dimensions, effectively degenerating into standard BO within a large search space and yielding weak local modeling. These issues arise from the reliance of BAxUS on a failure-count-based mechanism to control trust-region shrinkage and subspace expansion, where the threshold $\tau_{\text{fail}}$ is often too small in low dimensions and too large in high dimensions. We therefore outline a budgeted nested-subspace schedule that promotes thorough early exploration and more reliable scaling.

**Sample-Efficient 1D Initialization.** To mitigate early over-expansion, GRABBO starts the optimization process in a fixed 1-dimensional subspace ($d_1 = 1$), rather than calculating an initial dimension $d_1$ such that the final dimension after exponential expansion would be as close as possible to the original problem dimension. This design is motivated by the observation that the difficulty of optimization grows exponentially with problem dimensionality. By projecting the full space into

a single dimension, we can obtain a coarse yet informative estimate of promising regions with only a small number of evaluations. Since subspace-based BO tends to search around the current best solution, a strong initialization significantly improves downstream optimization performance.

**Budgeted subspace expansion.**   Rather than relying on failure counts, GRABBO allocates a fixed evaluation budget to each subspace and expands only once this budget is exhausted. The fixed evaluation budget for each subspace is determined by a principled strategy based on regret, which in this context represents the gap between the best value found so far and the true global optimum. The specifics of this allocation strategy are detailed in Section 4.1. This strategy prevents premature dimensional increases arising from stochastic failures and ensures that early-stage subspaces are thoroughly explored before advancing to higher dimensions. It also decouples the subspace expansion logic from noisy local behavior, improving robustness and reproducibility.

**Effective vs ambient dimension.**   The expansion strategy in GRABBO is guided by the assumed complexity of the objective, which hinges on the relationship between its effective dimension, $d_e$, and the ambient dimension, $D$. For simple objectives, $d_e$ is presumed to be small, with the high $D$ resulting primarily from many irrelevant features. In this regime, expanding to the full space is unnecessary, as the probability of multiple $d_e$ being projected into the same subspace coordinate is low. Conversely, for complex objectives where $d_e$ may be large or unknown, allowing expansion towards the full space can serve as a crucial safeguard, ensuring that all variables are eventually considered and preventing the premature restriction of the search space.

**Practical Safeguard of Maximum Expansion.**   Full expansion to the ambient dimension $D$ can become impractical for objectives with several thousand or more dimensions. Such scales exceed the practical capacity of standard Bayesian Optimization (Frazier, 2018) and can render trust-region mechanisms ineffective. To provide a practical safeguard against this, GRABBO imposes an upper bound $D^*$ on the final subspace dimension. As we discuss in Section 3.2 and have observed empirically, once the dimensionality approaches 1000, the trust region rarely shrinks. Based on this finding, we set $D^* = 1024$ as a default. We have found this value works well across a variety of tasks, though practitioners may still specify a different $D^*$ based on domain knowledge.

## 4.1 Heuristic Regret-Based Allocation under Nested Subspace Expansion

We consider the BAxUS optimization process as a multi-stage Bayesian optimization procedure. At each stage $i$, optimization is performed within a $d_i$-dimensional subspace using $n_i$ function evaluations. The total budget is constrained by $\sum_{i=1}^{k} n_i = N$. The dimensionality expands across stages, with $d_i = d_1 b^{i-1}$ for some base $b > 1$, culminating in the full-dimensional space $d_k = D$.

We derive a principled heuristic for allocating the evaluation budget $\{n_i\}$ across these stages by modeling the cumulative regret. In the subspace setting, the true cumulative regret at stage $i$ is the sum of the cumulative subspace regret and the cumulative approximation regret. The cumulative subspace regret $R_{n_i,d_i}$ measures how much the function values obtained over the $n_i$ evaluations deviate from the optimal value within the $d_i$-dimensional subspace. The cumulative approximation regret $\Delta_{d_i}$ captures the gap between this subspace optimum and the true global optimum in the full space. Allocating $\{n_i\}$ appropriately is crucial for balancing the trade-off between reducing cumulative subspace regret and reducing cumulative approximation regret. Because our method uses a trust-region mechanism and therefore tends to exploit points near the current best solution, modeling cumulative regret better aligns budget selection $\{n_i\}$ with minimizing true regret and helps guide the search toward better solutions.

### 4.1.1 Modeling Subspace and Full-Space Regret

We begin by modeling the subspace cumulative regret $R_{n_i,d_i}$ from optimizing a smooth objective function using $n_i$ queries within a $d_i$-dimensional subspace. Under standard assumptions for Gaussian Process Thompson Sampling with squared exponential (RBF) kernels (Chowdhury & Gopalan, 2017), the cumulative subspace regret satisfies:

$$R_{n_i,d_i} = \widetilde{\mathcal{O}}\left(\gamma_{n_i,d_i}\sqrt{n_i d_i}\right), \quad \text{where} \quad \gamma_{n_i,d_i} = \mathcal{O}\left((\ln n_i)^{d_i+1}\right). \tag{1}$$

This captures the cost of searching within the subspace assuming the true optimum lies in it. However, this does not reflect the true regret when the global optimum lies outside the subspace. To

model this, we consider the regret at a per-step level. The average per-step regret within the subspace is $r_{n_i,d_i} = R_{n_i,d_i}/n_i$. The true per-step regret must also account for the optimality gap, $\Delta_{d_i} := f_{d_i}^* - f_D^* \geq 0$, which is the penalty incurred at each step for operating in a subspace. Thus, the true per-step regret is $r_{n_i,d_i}^* = r_{n_i,d_i} + \Delta_{d_i}$, where $f_{d_i}^*$ denotes the optimal solution in the subspace with dimension $d_i$, and $f_D^*$ denotes the global optimal solution.

To account for the mismatch between the subspace dimension $d_i$ and the full ambient dimension $D$, we draw inspiration from the spectral truncation theory of smooth kernels. Belkin (Belkin, 2018) showed that for functions in the RKHS of a smooth radial kernel, the approximation error ($\Delta_{d_i}$) from projecting onto a finite-dimensional subspace decays nearly exponentially with the number of retained dimensions. This theoretical insight implies that smooth functions are highly compressible, and discarding $(D - d_i)$ dimensions induces an exponentially small loss in captured information.

Motivated by this property, we model the optimality gap with a heuristic exponential penalty, approximating per-step regret by replacing the additive gap with a multiplicative factor:

$$r_{n_i,d_i}^* \approx C \cdot r_{n_i,d_i} \cdot \exp(\alpha(D - d_i)). \tag{2}$$

The cumulative true regret $R_{n_i,d_i}^*$ over $n_i$ steps is obtained by multiplying the average per-step regret by $n_i$. By substituting our heuristic for $r_{n_i,d_i}^*$ and the definition $r_{n_i,d_i} = R_{n_i,d_i}/n_i$, we derive the approximation for $R_{n_i,d_i}^*$:

$$R_{n_i,d_i}^* = n_i \cdot r_{n_i,d_i}^* \approx n_i \cdot (C \cdot r_{n_i,d_i} \cdot \exp(\alpha(D - d_i))) = C \cdot R_{n_i,d_i} \cdot \exp(\alpha(D - d_i)), \tag{3}$$

where $\alpha > 0$ and $C$ are abstract constants, rather than tunable parameters. This multiplicative form, derived from a per-step analysis, retains the essential trade-off implied by spectral truncation theory while ensuring analytical tractability. This design ensures that the true cumulative regret exhibits exponential decay with respect to $d_i$, and it also guarantees that when $d_i$ expands to $D$, the formulation recovers the full-space cumulative regret. Substituting the expression for $R_{n_i,d_i}$ yields:

$$R_{n_i,d_i}^* = \widetilde{\mathcal{O}}\left((\ln n_i)^{d_i+1} \cdot \sqrt{n_i d_i} \cdot \exp(\alpha(D - d_i))\right). \tag{4}$$

We now rewrite the polylogarithmic term in exponential form:

$$(\ln n_i)^{d_i+1} = \exp\left((d_i + 1)\ln\ln n_i\right),$$

and combine it with the penalty to obtain:

$$R_{n_i,d_i}^* = \widetilde{\mathcal{O}}\left(\sqrt{n_i d_i} \cdot \exp\left((d_i + 1)\ln\ln n_i + \alpha(D - d_i)\right)\right). \tag{5}$$

This expression reveals a fundamental trade-off: increasing $d_i$ reduces the approximation error exponentially but increases the cumulative subspace regret. Since the exact expression with exponential terms is analytically intractable, we adopt the simplifying assumption that these opposing exponential effects approximately balance out: $\exp\left((d_i + 1)\ln\ln n_i + \alpha(D - d_i)\right) \approx \mathcal{O}(1)$, yielding a tractable surrogate regret expression:

$$R_{n_i,d_i}^* = \widetilde{\mathcal{O}}\left(\sqrt{n_i d_i}\right). \tag{6}$$

This is not a formal bound, but a heuristic motivated by the exponential decay property in spectral truncation theory, designed to capture the key trade-off budget allocation across nested subspaces.

### 4.1.2 OPTIMAL BUDGET ALLOCATION

A seemingly natural method is to directly minimize the total regret across stages, but this approach has an inherent flaw originating from its mathematical form. Because the regret expression inherits concavity from the square root function, the optimization process tends to produce a type of extreme allocation known as a corner solution. In such a solution, the budget becomes highly concentrated on a few stages while the remaining ones receive virtually no resources. A corner solution is undesirable because it is not a sensible strategy, but merely an inevitable artifact of minimizing a concave function, which favors values on the boundary of the feasible region. This fundamentally prevents a balanced and smooth allocation of resources.

To address this issue, we reformulate the allocation task by balancing approximate marginal regrets across stages. The marginal regret at stage $i$, derived from Eq. equation 6 as the decrease in regret with respect to $n_i$, is given by $\Delta_i = \frac{\sqrt{d_i}}{2\sqrt{n_i}}$.

Instead of minimizing the total regret, we minimize the maximum marginal regret:

$$\min_{\{n_i\}} \max_{i=1,\dots,k} \frac{\sqrt{d_i}}{2\sqrt{n_i}} \quad \text{subject to} \quad \sum_{i=1}^{k} n_i = N, \ \ n_i \geq 0. \tag{7}$$

This min–max program directly acts on the marginal regrets. Since $1/\sqrt{n_i}$ is convex and strictly decreasing, the optimum is characterized by equalizing the marginal regrets across all stages, i.e. $\frac{\sqrt{d_i}}{2\sqrt{n_i}} = \mu$ for some constant $\mu > 0$. Solving for $n_i$ gives $n_i = d_i/(4\mu^2)$, which implies that the allocation is proportional to the subspace dimensionality, i.e. $n_i \propto d_i$. Enforcing the budget constraint $\sum_{i=1}^{k} n_i = N$ then yields the closed-form solution

$$n_i = \frac{d_i}{\sum_{j=1}^{k} d_j} N. \tag{8}$$

This reformulation provides a principled and interpretable allocation rule. By equalizing the active $\Delta_i$, it ensures balanced approximate marginal regrets across stages, directly controls the worst-case stage, and prevents pathological budget concentration—leading to a robust distribution of resources.

However, in practical Bayesian optimization, the evaluation budget $N$ is often insufficient to locate the global optimum. As a consequence, directly applying the proportional allocation $n_i \propto d_i$ may cause under-exploration in early low-dimensional subspaces. To address this issue, we introduce a uniform baseline across all stages. Specifically, we reserve an $\eta$ fraction of the total budget and distribute it evenly across the $k$ subspaces, while allocating the remaining $(1 - \eta)N$ proportionally to dimensionality:

$$n_i = \frac{\eta N}{k} + (1 - \eta) \cdot \frac{d_i}{\sum_{j=1}^{k} d_j} \cdot N. \tag{9}$$

This hybrid allocation guarantees each subspace a minimum number of evaluations for meaningful exploration, while respecting the dimension-proportional principle from the surrogate regret model.

### 4.2 PSEUDOCODE OF GRABBO

Algorithm 1 details the execution flow of GRABBO. GRABBO takes as input the expansion base $b$, ambient dimensionality $D$, maximum target dimension $D^\star$, total budget $N$, number of initial samples $n_{\text{init}}$, budget-allocation fraction $\eta$, number of full trust-region contractions $\gamma$, and a boolean flag Expanded. GRABBO first computes the number of stages, $k$, and constructs an embedding matrix $S_i^\top$ for each $d_i$-dimensional stage. Before the main optimization loop, GRABBO pre-computes for each stage $i \in 1, \dots, k$: 1) the budget $n_i$, 2) the failure tolerance counter$_i$, and 3) the cumulative budget threshold (total evaluation count triggering a transition), $n_i^{\text{cum}}$.

---

**Algorithm 1: GRABBO Algorithm**

**Require:** $b, D, D^\star, N, n_{\text{init}}, \eta, \gamma$, Expanded
**Ensure:** $x^\star \in \arg\min_{x \in \mathcal{X}} f(x)$

  $D \leftarrow \min\{D, D^\star\}$
  $k \leftarrow \lceil \log_b D \rceil$
  **for** $i = 1, \dots, k$ **do**
    $d_i \leftarrow \min\{b^{i-1}, D\}$
  **end for**
  Construct embeddings $S_i^\top : \mathbb{R}^{d_i} \to \mathbb{R}^D$
  Sample $n_{\text{init}}$ initial points in $\mathbb{R}^{d_1}$
  Evaluate, and fit GP surrogate
  $k' \leftarrow k$ **if** Expanded **else** $k - 1$
  **for** $i = 1, \dots, k'$ **do**
    $n_i \leftarrow \left\lfloor \frac{\eta(N - n_{\text{init}})}{k'} + \frac{(1-\eta)d_i}{\sum_{j=1}^{k'} d_j}(N - n_{\text{init}}) \right\rfloor$
    counter$_i \leftarrow \max(1, \lfloor \frac{n_i}{2\gamma} \rfloor)$
  **end for**
  $n_i^{\text{cum}} \leftarrow n_{\text{init}} + \sum_{j=1}^{i} n_j$
  $L \leftarrow L_{\text{init}}$; stage $\leftarrow 1$
  Initialize success and failure counters
  **for** $t = n_{\text{init}}, \dots, N - 1$ **do**
    **if** $t \geq n_{\text{stage}}^{\text{cum}}$ **and** stage $< k'$ **then**
      stage $\leftarrow$ stage $+ 1$
      Update $S^\top$ to dimension $d_{\text{stage}}$
      Reset counters and $L$
    **end if**
    Adjust TR size $L$ and counters ▷ See in 4.2
    Find $y$ in current subspace $\mathbb{R}^{d_{\text{stage}}}$ within TR
    and evaluate $f(S_{\text{stage}}^\top y)$
    Add new data point to dataset and re-fit GP
  **end for**
  **Return:** $\arg\min_{(S^\top y, f(S^\top y)) \in \mathcal{D}} f(S^\top y)$

---

The main optimization loop then commences, tracking its progress with an internal 'stage' variable that starts at one. The core novelty of the algorithm is its budget-driven expansion schedule, where it transitions to the next higher-dimensional subspace by incrementing the 'stage' variable as soon as the total evaluation count $t$ meets or exceeds the cumulative budget threshold for the current stage, $n_{\text{stage}}^{\text{cum}}$. Within any given stage, the algorithm dynamically adjusts the trust region (TR) side length $L$ in a manner consistent with BAxUS (see A for details).

## 5 EXPERIMENTS

We evaluate our proposed GRABBO algorithm on eight benchmarks, consisting of three synthetic functions and five real-world tasks.

**Benchmarks.** For the synthetic benchmarks, we select the commonly used Branin, Levy, and Hartmann functions with effective dimensionalities $2, 4$, and $6$, respectively. Following Wang et al. (2016), we construct high-dimensional variants of the synthetic functions by appending dummy variables that do not influence the function values. Specifically, we expand Branin to 500 dimensions, and Levy and Hartmann to 1000 dimensions. For Levy, we follow the input domain setting proposed by Hvarfner et al. (2024), which avoids search directions aligned with the diagonals of the space and prevents the optimizer from gaining an unreasonably large advantage. For the real-world tasks, we considered three LASSO benchmarks (Šehić et al., 2022): Lasso-DNA (180D), Lasso-Leukemia (7129D), and Lasso-RCV1 (47236D), and two MuJoCo control benchmarks: HalfCheetah-v4 (102D) (Wawrzyński, 2009) and Humanoid-v4 (6392D) (Tassa et al., 2012). The evaluation budgets follow prior work (Šehić et al., 2022; Hvarfner et al., 2024; Xu et al., 2024; Ngo et al., 2025): 1000 evaluations for the synthetic functions, 300 for Lasso-DNA, and 1000 for Lasso-Leukemia, Lasso-RCV1, HalfCheetah and Humanoid. To match the minimization setting of GRABBO, we negate HalfCheetah and Humanoid.

**Experimental Setup.** We compare against six representative HDBO methods covering different paradigms: subspace-based (BOIDS (Ngo et al., 2025), BAxUS (Papenmeier et al., 2022b), HeSBO (Nayebi et al., 2019)), trust-region (TuRBO-1 (Eriksson et al., 2019)), and standard BO variants (VBO (Hvarfner et al., 2024), SBO-Matérn (RI), abbreviated as SBO (Xu et al., 2024)). The detailed implementations of all baselines as well as GRABBO are deferred to Appendix B.1. All optimizers are initialized with 10 samples, except for the Humanoid task where we use 30 initial samples. For the three synthetic tasks, where the true global optimum is known, we report the regret. For the five real-world tasks where the true optimum is unknown, we report the best minimum value found during the optimization process. We ran each experiment 10 times independently and report the mean $\pm$ one standard error across runs. Additional ablation and hyperparameter sensitivity experiments are provided in Appendix C and D, respectively.

**Synthetic Benchmark Results.** Figure 1 (a–c) summarizes the performance of our method compared with baselines on three synthetic benchmarks. GRABBO achieves the best performance across all three synthetic benchmarks. Notably, on the latter two functions, which involve higher intrinsic dimensionality, GRABBO outperforms all baselines by more than an order of magnitude in final regret. While BOIDS performs comparably to GRABBO on Branin, its optimization nearly stalled on Hartmann and Levy, showing progress comparable to TuRBO, and failing to effectively exploit the higher-dimensional structure.

BAxUS, SBO, and VBO exhibit similar behavior on these synthetic benchmarks, each showing a tendency to converge prematurely to suboptimal regions in certain tasks. For example, on the Levy function, BAxUS progresses slowly, whereas VBO plateaued during the middle stage of optimization but occasionally discovered better values later, resulting in a sudden drop in regret and overall better final performance. SBO, while initially strong on the Hartmann benchmark, is unable to make further improvements after approximately 400 evaluations.

**Real-World Benchmark Results.** Results on real-world tasks are shown in Figure 1 (d–h). On Lasso-DNA, GRABBO achieved strong initialization and stable optimization. Although its convergence is slower, it ultimately found the best solution with significantly lower variance than VBO and SBO. This may reflect the moderate ambient dimension of the task and its relatively high effective dimension, where subspace-based methods provide limited benefits. While VBO and SBO barely improved (final regret around 0.37), both GRABBO and BAxUS start with good initialization. GRABBO quickly reached 0.2 within 200 evaluations and further converged to 0.15, outperforming all baselines. BOIDS runs out of memory after 500 steps. On Lasso-RCV1, the most difficult task, VBO and SBO stagnate after 300 steps around 0.26. TuRBO can not run due to out-of-memory (Šehić et al., 2022). BAxUS expands to the full space after 170 steps, leading to prohibitive runtime and an incomplete run, while BOIDS also fails due to memory limits, with detailed failure modes for the baselines provided in Appendix B.4. In contrast, GRABBO maintains steady progress despite the extreme dimensionality, avoided stagnation, and achieved the best final performance.

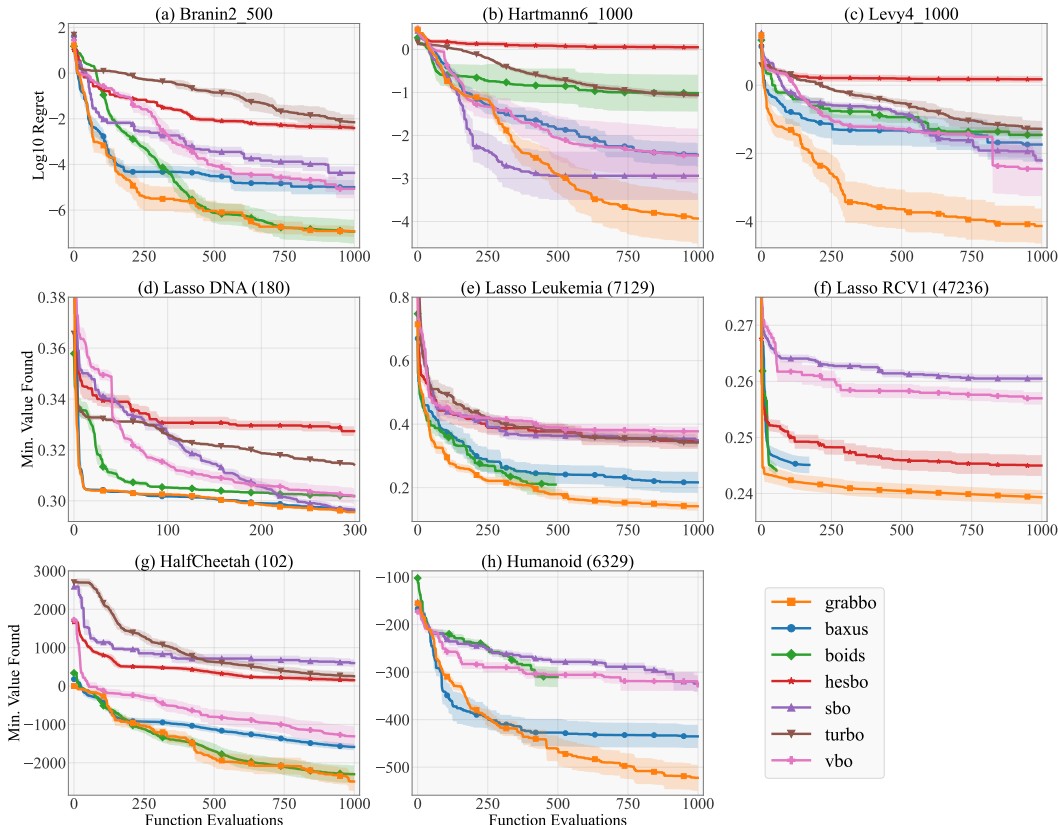

Figure 1: Optimization performance on synthetic (a-c) and real-world (d-h) benchmarks.

Compared to the Lasso benchmarks, the MuJoCo tasks likely involve higher effective dimensionality, which would intuitively pose a greater challenge for subspace-based methods. However, we observe that these methods perform even more effectively in the MuJoCo setting. On the HalfCheetah benchmark, both GRABBO and BOIDS achieve similarly strong performance, successfully identifying solutions with objective values below $-2000$. BAxUS and VBO follow with slightly weaker results, and both of these methods still perform substantially better than SBO. A similar trend is observed on the Humanoid benchmark. GRABBO and BAxUS are able to consistently discover better solutions throughout the optimization process, while VBO and SBO demonstrated limited progress. BOIDS, once again, fails to complete the run due to memory exhaustion. These results suggest that, in certain high-dimensional environments such as those found in MuJoCo tasks, subspace methods can remain effective. GRABBO, in particular, shows strong and stable performance across both tasks, even in the presence of large-scale search spaces.

## 6 CONCLUSIONS

We propose GRABBO, a regret-aware subspace Bayesian optimization framework that effectively balances exploration within each subspace and controlled expansion to higher dimensions. Our analysis reveals that thorough exploration of nested low-dimensional subspaces is critical under the subspace assumption, as it provides reliable guidance for subsequent expansions. Furthermore, although projecting multiple effective dimensions onto the same subspace coordinate may impair optimization, our experimental results show that the adverse effect is less pronounced than that caused by expanding into higher-dimensional spaces, where the subspace assumption breaks down and the method degenerates into standard BO prone to local optima. GRABBO avoids this failure mode and achieves robust performance across synthetic and real-world benchmarks.

## ETHICS STATEMENT

We affirm that this work adheres to the ICLR Code of Ethics . The research does not involve human subjects, personally identifiable information, or sensitive data. All datasets used are publicly available benchmark tasks (Branin, Hartmann, Levy, LassoBench, and MuJoCo environments), and we follow the licenses and intended use of these datasets. Our proposed method, GRABBO, is designed as a generic optimization algorithm and does not raise foreseeable risks of harmful applications or ethical concerns beyond those already present in Bayesian optimization research. We report experimental results faithfully, disclose all baselines and hyperparameters, and ensure compliance with research integrity and reproducibility standards.

## REPRODUCIBILITY STATEMENT

We provide anonymized source code in the supplementary materials. The current version has not been fully organized, but it is runnable and demonstrates our proposed method. All datasets used in our study are standard and publicly available benchmarks (Branin, Hartmann, Levy, LassoBench, and MuJoCo environments). To aid reproducibility, we include algorithmic descriptions and pseudocode in Section 4, benchmark specifications and hyperparameter settings in Section 5 and Appendix B, and additional ablation and sensitivity analyses in Appendices C and D. After publication, we will release a fully organized and documented codebase that enables reproduction of all results.

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

## A    GRABBO TRUST REGION DYNAMICS

As discussed in 4.2, GRABBO utilizes a trust region that is dynamically adjusted during optimization. Following BAxUS (Papenmeier et al., 2022a), GRABBO updates the length $L$ of the trust region using internal success and a failure counters. If an evaluation finds a better solution, the success counter is incremented and the failure counter is reset; otherwise, the failure counter is incremented and the success counter is reset. The TR side length $L$ is doubled if the success counter reaches a small, predefined threshold, and it is halved if the failure counter reaches the stage-specific tolerance counter$_i$. Following any adjustment, the corresponding counter is reset to prepare for the next sequence of evaluations.

## B    EXPERIMENTAL DETAILS

### B.1    EXPERIMENTAL METHODS

For GRABBO, we set $b = 4$, using `Expanded=False` and $\eta = 5\%$ on synthetic benchmarks, and `Expanded=True` with $\eta = 5\%$ on real-world tasks. We set the maximum dimensionality at 1024 for all the real-world benchmarks.

As for baselines, we compare our proposed method against a diverse set of state-of-the-art high-dimensional Bayesian optimization algorithms:

- HeSBO. The implementation we adopt is from the official repository released by the authors: `https://github.com/aminmnayebi/HesBO`.
- TuRBO. For TuRBO, we relied on the original open-source implementation published by Uber AI Labs: `https://github.com/uber-research/TuRBO`.
- BAxUS. Our experiments use the public codebase maintained by the BAxUS authors: `https://github.com/LeoIV/BAxUS`.
- VBO. The code for Vanilla Bayesian Optimization Performs Great in High Dimensions is available at: `https://github.com/XZT008/Vanilla_BO_in_High_D`.
- SBO (SBO-Matérn (RI)). We employed the official implementation of Standard gaussian process is all you need for high-dimensional bayesian optimization provided here: `https://github.com/XZT008/Standard-GP-is-all-you-need-for-HDBO`.
- BOIDS. The experiments for BOIDS were run using the repository published by the authors: `https://github.com/LamNgo1/boids`.

### B.2    BENCHMARKS

Our experiments cover both synthetic benchmark functions and real-world applications to comprehensively evaluate algorithm performance.

**Synthetic functions.**    We adopt several well-known continuous optimization test functions widely used in the BO literature:

- **Branin (2D).** The Branin function is defined on $[-5, 10] \times [0, 15]$ as
$$f(x) = a(x_2 - bx_1^2 + cx_1 - r)^2 + s(1 - t)\cos(x_1) + s,$$
where $a = 1$, $b = \frac{5.1}{4\pi^2}$, $c = \frac{5}{\pi}$, $r = 6$, $s = 10$, and $t = \frac{1}{8\pi}$. It has three global optima with $f(x^\dagger) \approx 0.3979$ located at $(-\pi, 12.275)$, $(\pi, 2.275)$, and $(9.42478, 2.475)$.
- **Hartmann (6D).** The 6-dimensional Hartmann function is given by
$$f(x) = -\sum_{i=1}^{4} \alpha_i \exp\left(-\sum_{j=1}^{6} A_{ij}(x_j - P_{ij})^2\right),$$
where $x \in [0, 1]^6$, and matrices $A = [A_{ij}]$, $P = [P_{ij}]$, and coefficients $\alpha_i$ are predefined constants. The global optimum is $f(x^\dagger) \approx -3.32227$ at $x^\dagger \approx (0.2017, 0.1500, 0.4769, 0.2753, 0.3117, 0.6573)$.

- **Levy (4D).** We use a 4-dimensional variant of the Levy function with effective dimensionality $d = 4$. It is defined as

$$f(x) = \sin^2(\pi w_1) + \sum_{i=1}^{d-1}(w_i - 1)^2 \left[1 + 10\sin^2(\pi w_i + 1)\right] + (w_d - 1)^2 \left[1 + \sin^2(2\pi w_d)\right],$$

  where $w_i = 1 + \frac{1}{4}(x_i - 1)$. The search space is

  $$[-10, 5] \times [-10, 10] \times [-5, 10] \times [-1, 10],$$

  and the global minimum is $f(x^\dagger) = 0$ attained at $x^\dagger = (1, 1, 1, 1)$.

**Real-world tasks.** We also evaluate on several high-dimensional real-world applications:

- **LASSO-DNA.** A regression task based on a genomics dataset, where the objective is the cross-validation error of an $\ell_1$-regularized linear regression model. The dimensionality is $d = 180$. The dataset is taken from the LassoBench benchmark suite.

- **LASSO-Leukemia.** Another genomics regression problem on leukemia expression data with dimension $d = 7129$. As in Lasso-DNA, the optimization task is to tune LASSO hyperparameters to minimize prediction error. This dataset is also included in LassoBench.

- **LASSO-RCV1.** A text classification dataset derived from the RCV1 corpus, represented as a sparse bag-of-words feature space. In the LassoBench paper (Šehić et al., 2022), the dimensionality is reported as $d = 19959$. However, the RCV1-v2 corpus used in the official LassoBench code actually has $d = 47236$ features (Lewis et al., 2004), and when directly running the released code the dimensionality is confirmed to be $d = 47236$. Therefore, we follow the code implementation and treat the dataset as $d = 47236$-dimensional, with the optimization objective being the cross-validation error of the LASSO model.

- **HalfCheetah.** A reinforcement learning control task in the Mujoco simulator. We optimize policy parameters of dimensionality $d = 102$ for the HalfCheetah-v4 environment. The objective is to maximize episode return under the learned policy. The implementation follows the Mujoco benchmark suite.

- **Humanoid.** A challenging locomotion task in the Mujoco simulator, where a humanoid robot must learn to walk and balance. The policy search space is high-dimensional, with $d = 6392$ parameters for the Humanoid-v4 environment. The optimization objective is to achieve the highest possible episode return, reflecting stable and efficient locomotion. Our setup is consistent with the standard Mujoco control benchmarks.

### B.3 EXPERIMENTAL SETUP

In all of our main experiments, we fix the algorithmic parameters of GRABBO as follows. The parameter $\eta$ denotes the proportion of the total evaluation budget that is uniformly distributed across all subspaces in addition to the linearly allocated evaluations. For example, if the total budget is $N$ and there are $K$ subspaces, then $\eta N$ evaluations are evenly divided among all $K$ subspaces, while the remaining $(1 - \eta)N$ evaluations follow the linear allocation rule. We set $\eta = 0.05$ in our experiments.

The parameter $b$ controls the dimensionality expansion rule: when a subspace is expanded, each existing dimension in the subspace is replaced by $b$ new dimensions. In other words, one old dimension is split into $b$ new dimensions, so the size of the subspace scales multiplicatively with $b$ at each expansion step. We set $b = 4$ in our experiments.

For the trust region mechanism, the base length parameters are: initial $0.8$, maximum $1.6$, and minimum $0.0078125$. The acquisition function is Thompson Sampling, and the surrogate model is a Gaussian process with a Matérn-$3/2$ kernel and Automatic Relevance Determination (ARD) lengthscales.

For SBO and VBO, since their initialization relies on Sobol sequences, which are limited to a maximum dimension of 21201, we modified their initialization method to uniform sampling for the Lasso-RCV1 benchmark, whose dimensionality exceeds this Sobol limit.

B.4   REASONS FOR THE SCALABILITY FAILURE OF BASELINE METHODS

We ran the BOiDS experiments on a workstation equipped with RTX 3080 GPUs (20GB VRAM), 12 vCPUs (Intel Xeon Platinum 8352V @ 2.10GHz), and 48GB of system memory for the Lasso-Leukemia, Lasso-RCV1, and Humanoid benchmarks. For BAxUS, we used the same workstation with RTX 3080 GPUs on the Lasso-Leukemia and Lasso-RCV1 benchmarks, while the Humanoid benchmark was run on an RTX A5000 GPU (24GB VRAM). Both methods encountered scalability issues when applied to ultra-high-dimensional benchmarks, either due to GPU memory exhaustion or excessive runtime.

In particular, BOIDS maintains multiple incumbent-guided search lines at each iteration and applies NSGA-II for multi-objective acquisition optimization along these lines. While effective in lower-dimensional settings, this design leads to substantial memory overhead for GP batch prediction and candidate evaluation. On the over 6K-dimensional Humanoid, Lasso-Leukemia, and Lasso-RCV1 benchmarks, BOIDS consistently triggered out-of-memory errors. This suggests that its acquisition optimization pipeline cannot scale to ultra-high-dimensional domains within practical GPU constraints.

Similarly, the BAxUS method suffers from extreme runtime inefficiency in high dimensions. On the Lasso-RCV1 benchmark, we observe that around iteration 170, BAxUS expands the search subspace to the full ambient space. From this point onward, the method stalls entirely: no new evaluation is produced for over an hour. At this rate, completing the 1000-step optimization budget would require more than 30 days of continuous execution, which is clearly infeasible in practice. Moreover, the performance curve shows minimal improvement in the best-found value after expansion, indicating that BAxUS fails to make effective progress in this setting.

For the humanoid benchmark, we include only advanced methods such as BAxUS, VBO, BOiDS, and SBO as baselines. The rationale is that the work of Xu et al. (2024), which introduced SBO, conducted comparisons only with VBO on their self-constructed 1003-dimensional Humanoid-Standup subproblem. Based on this, we consider earlier methods insufficient for optimizing this problem and therefore exclude them from comparison. In addition, for BAxUS we report average results over only 4 runs on the humanoid benchmark, as after 400 evaluations, the BAxUS method slowed to completing only 6 evaluations per hour, and by around 700 evaluations, the speed further decreased to approximately 1 evaluation per hour. Such computational overhead is prohibitive within our available time budget. For the camera ready verison, we plan to complete and add six additional runs of BAxUS (allowing us to provide BAxUS results averaged over 10 total runs on this task).

Overall, these observations demonstrate that baseline methods relying on full GP updates or multi-objective subspace expansion face scalability limitations, both in terms of memory and wall-clock time, and cannot be applied reliably to optimization problems with thousands of dimensions.

## C   ALATION STUDIES

We perform ablation experiments on two representative and challenging benchmarks: the 1000-dimensional extension of the Hartmann6 function and the real-world Lasso-Leukemia task. The Hartmann6 function is a classical synthetic benchmark with intrinsically low effective dimensionality, but when embedded into 1000 dimensions it becomes extremely difficult and serves as a stress test for whether an algorithm can identify and exploit low-dimensional structure without being overwhelmed by the ambient dimensionality. In contrast, the Lasso-Leukemia task represents a real-world high-dimensional regression problem with noisy and correlated features, where the intrinsic structure is less explicit and the optimization landscape is substantially harder to navigate than in synthetic benchmarks. Together, these two problems highlight complementary challenges: Hartmann6-1000 emphasizes exploiting low effective dimensionality in a vast search space, while Lasso-Leukemia tests robustness under noisy, real-world conditions.

In Figure 2, we compare the following four variants to isolate the contribution of each component:

- **BAxUS init 2**: baseline BAxUS with the initial subspace dimension set to 2, serving as the unimproved reference method.

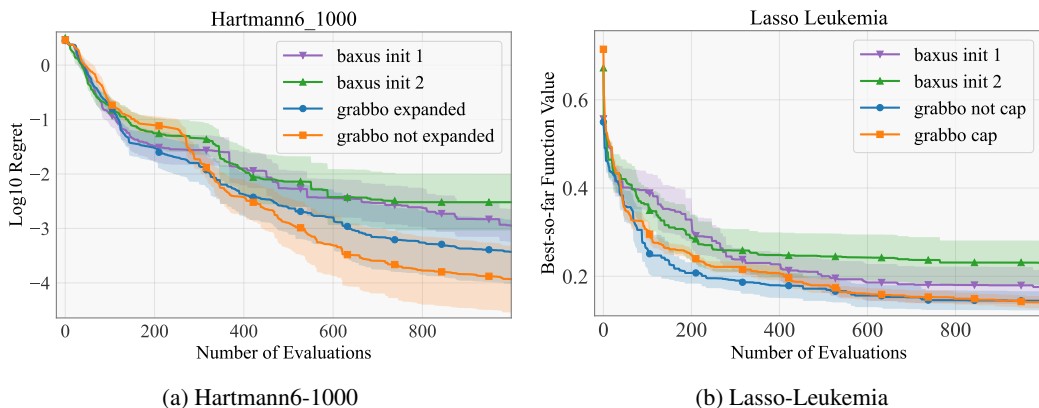

(a) Hartmann6-1000            (b) Lasso-Leukemia

Figure 2: Ablation study on Hartmann6-1000 and Lasso-Leukemia benchmarks.

- **BAxUS init 1**: a variant that reduces the initial subspace dimension to 1, emphasizing aggressive low-dimensional initialization.

- **GRABBO expanded/ not cap**: built upon BAxUS init 1, this variant introduces our proposed linear allocation of evaluation budgets across subspaces. The term "expanded / not cap" reflects benchmark-specific behavior: for Hartmann, the subspace is expanded until reaching the full ambient dimension; for Lasso-Leukemia, no maximum expansion cap is imposed.

- **GRABBO not expanded/ cap**: further modifies the above by avoiding full expansion. For Hartmann, the subspace stops short of the ambient dimension; for Lasso-Leukemia, a maximum cap of $1024$ dimensions is enforced.

The results show that on the Hartmann benchmark, all methods except *BAxUS init 2* reach errors around or below $10^{-3}$, surpassing the current state-of-the-art. Each modification—starting from dimension 1, applying linear allocation, and finally restricting full expansion—progressively improves performance. On the real-world Lasso-Leukemia benchmark, each modification also brings gains, though whether a cap is imposed has little effect, suggesting that low-dimensional subspace exploration itself already plays a key role in driving optimization in extremely high-dimensional problems.

## D   PARAMETER SENSITIVITY ANALYSIS

To further investigate the robustness of our method, we conduct a sensitivity analysis on the two key parameters introduced in our algorithm: $\eta$ and $b$. The parameter $\eta$ controls the additional budget compensation allocated to each subspace beyond the linear allocation rule, while $b$ determines the number of dimensions by which the search subspace is expanded at each stage. We perform these experiments on the same two representative benchmarks as in the ablation study, namely Hartmann6-1000 and Lasso-Leukemia. The results are shown on Figure 3.

For $\eta$, we consider five representative values: 0, 0.07, 0.05, 0.07, and 0.10. The case $\eta = 0$ corresponds to no additional compensation, meaning that the evaluation budget for each subspace is determined strictly by linear allocation. At the other extreme, $\eta = 0.10$ represents a setting where lower-dimensional subspaces receive a large amount of extra budget, thus emphasizing more thorough exploration in the early stages of the search. The intermediate value $\eta = 0.05$ provides a balanced trade-off between early exploration and later-stage refinement. We do not consider values of $\eta$ greater than $0.10$, since excessive compensation would starve higher-dimensional subspaces of evaluations, despite the fact that more complex structures typically emerge in higher dimensions. Our intention is to mitigate insufficient exploration in the lowest-dimensional subspaces, not to prioritize them exclusively.

For $b$, we vary the expansion factor across $\{3, 4, 5\}$. A value of $b = 1$ would imply no expansion at all, which defeats the purpose of progressive subspace growth, while $b = 2$ corresponds to a very

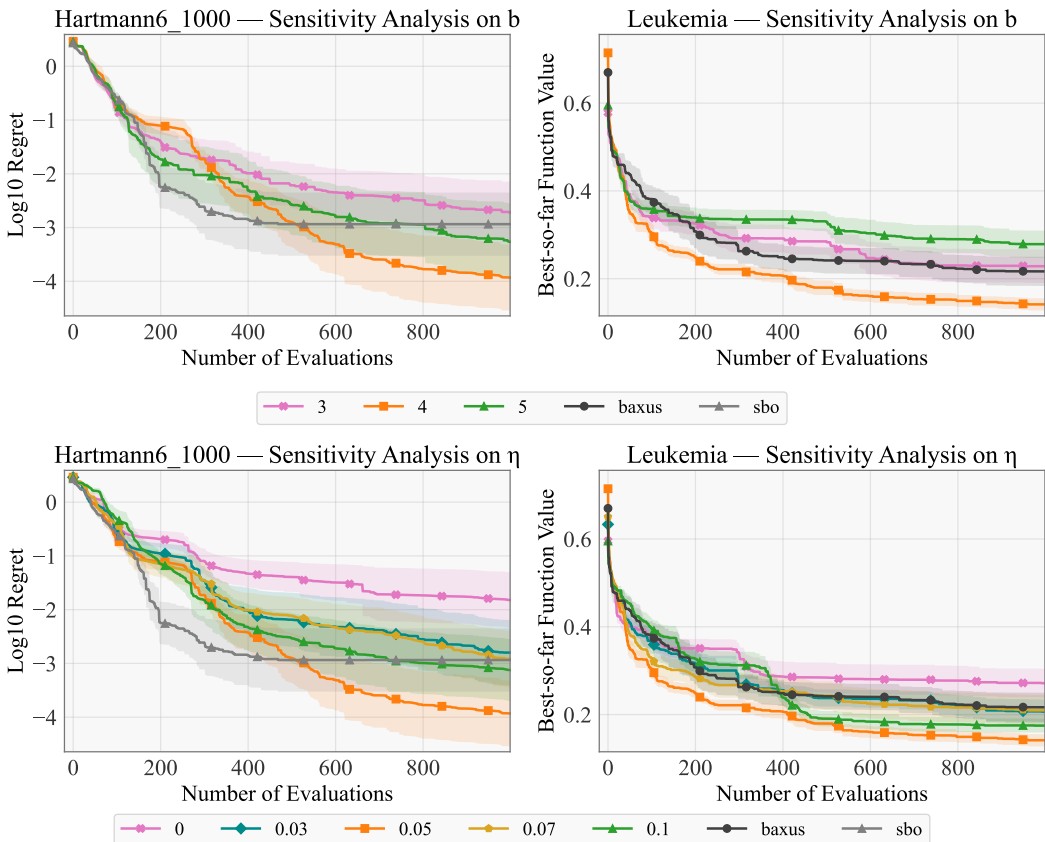

Figure 3: Parameter sensitivity analysis: (top) bin-based results and (bottom) eta-based results on Hartmann6 and Leukemia benchmarks.

slow expansion where the subspace dimension only doubles each time. Such slow growth leads to an excessive number of expansion stages, which can hinder effective low-dimensional subspace exploration. On the other hand, $b = 6$ would correspond to overly aggressive exponential expansion that rapidly escalates to the full ambient space. Hence, we restrict our study to the range $b \in [3, 5]$, which provides a meaningful balance between gradual subspace growth and timely access to higher-dimensional information.

The sensitivity analysis indicates that setting $b = 4$ is most effective, as it provides a balanced expansion rate. In addition, we capped the maximum dimensionality at 1024, which is divisible by 4, to avoid uneven allocation of evaluation budgets across subspaces that would otherwise result from the dimensionality limit. For $\eta$, we observe that all settings except $\eta = 0$, which corresponds to purely linear allocation of evaluations with respect to dimensionality, achieve performance close to or better than the strongest baseline. This highlights the importance of incorporating additional uniform allocation in order to enhance optimization performance.

# E    LLM USAGE STATEMENT

Large language models (LLMs) were used only as auxiliary tools to improve the writing and presentation of this paper and to assist in code implementation (e.g., debugging and code formatting). They were not involved in the generation or refinement of research ideas, experimental design, or theoretical developments. The authors take full responsibility for the entire content and the correctness of the code.

