# OpenReview forum: "Making the Subspace Assumption Work: Low-Dimensional Exploration for High-Dimensional Bayesian Optimization"
_ICLR.cc/2026/Conference — ICLR 2026 Conference Withdrawn Submission_

### Official Review · Reviewer_M7GV · 2025-10-24

**Soundness:** 3
**Presentation:** 2
**Contribution:** 2
**Rating:** 4
**Confidence:** 5

**Summary:**

This paper addresses the problem of high-dimensional black-box optimization. Various approaches have used Bayesian optimization (BO) for this problem type, often with additional assumptions on the structure of the objective function. One popular line of research models the surrogate model in a low-dimensional subspace and projects a candidate solution to the full-dimensional search space to evaluate it. Since fixing the dimensionality of the low-dimensional subspace is often overly strict, BAxUS introduced an expanding subspace that grows the dimensionality of the search space over time. This paper follows up on BAxUS by identifying shortcomings in several design decisions and proposing solutions to these shortcomings. In particular, the authors argue that BAxUS often expands early, low-dimensional subspace prematurely and shrinks high-dimensional subspaces too slowly. According to the authors, this is due to the mechanism by which BAxUS controls subspace expansion and trust region growth/shrinkage, which is based on the number of times the algorithm fails to consecutively find a better solution.

In contrast to BAxUS, they assign a fixed evaluation budget to each subspace, which is based on the expected simple regret in subspace-based  Bayesian optimization.

The authors evaluate the proposed algorithm on various synthetic and real-world scenarios, showing that the proposed approach can outperform the state-of-the-art.

**Strengths:**

The approach is clearly motivated. The design decisions are sound and empirically shown to be effective. The empirical evaluation is sound and features relevant state-of-the-art methods.

**Weaknesses:**

The overall innovation of BAxUS is limited. The main difference to BAxUS is the way the budget for a subspace is calculated. Other differences are 1) that GRABBO always starts with the lowest possible initial dimensionality and 2) that GRABBO can expand the subspace even if the trust region has not yet reached its minimum base length. It is worth mentioning that “Papenmeier, Leonard, Luigi Nardi, and Matthias Poloczek. ‘Bounce: Reliable high-dimensional Bayesian optimization for combinatorial and mixed spaces.’ Advances in Neural Information Processing Systems 36 (2023) also uses a fixed budget per subspace. Furthermore, Bounce outperforms BAxUS on continuous problems, so it might be worth benchmarking against it.
The fact that the paper features a hallucinated citation (Papenmeier, Daxberger, Rasmussen: “Baxus: Budget-aware subspace search for high-dimensional Bayesian optimization”) makes me question the author’s scientific rigor.
Minor comments:
-	Hvarfner et al., “Vanilla BO performs great in high dimensions” has been accepted to ICML

**Questions:**

-	In lines 93,94, you talk about challenges in kernel estimation. What exactly are those challenges?
-	Is it correct that a subspace might not “see” every trust region base length? Do you see a problem with that?

**Details Of Ethics Concerns:**

As written in my review, one reference is hallucinated.

---

### Official Review · Reviewer_9872 · 2025-10-30

**Soundness:** 2
**Presentation:** 2
**Contribution:** 2
**Rating:** 2
**Confidence:** 3

**Summary:**

This paper proposes GRABBO, a regret-aware budget allocation scheme for progressive subspace Bayesian optimization. The authors reformulate the subspace growth of BAxUS into a multi-subspace resource allocation problem: using a surrogate regret model, the method (heuristically) equalizes marginal regret across stages, which yields the rule $n_i \propto d_i$; and a small uniform fraction $\eta$ is added as a robustness baseline so every stage receives a minimum budget. The algorithm starts in 1D, expands by a factor $b$ using nested sparse embeddings, and runs BO within each subspace via a trust-region GP with Thompson sampling. Experiments cover synthetic and real benchmarks with ablations for subspace growth and sensitivity to $b,\eta$.

**Strengths:**

1. The paper recasts progressive subspace Bayesian optimization as a multi-stage budget allocation problem, which makes expansion decisions explicit and comparable across methods.
2. From the surrogate regret and marginal-regret equalization, the paper arrives at a proportional allocation $n_i \propto d_i$. This is easy to implement and reason about, unlike many Bayesian optimization heuristics.
3. The embedding design preserves previous observations across expansions, which can avoid data wastage and stabilize the GP fit as the dimension grows.

**Weaknesses:**

1. The surrogate regret (and the resulting proportional rule) hinges on an approximation; there is no bound on the approximation error or conditions under which it is accurate.
2. Budgets are chosen per stage as if stages were independent, but later stages reuse all earlier observations. In this case, $n_i\propto d_i$ rule can be misaligned with the true marginal value of evaluations.
3. The derivation leans on RBF kernel regret behavior, while experiments use a Matern-3/2 kernel (Line 752). The paper does not reconcile the implications of this mismatch.
4. The 1D start, fixed growth factor $b$, and a hard cap $D^\star=1024$ are practical but lack adaptability. There is a lack of data-driven decision criteria for "when to expand and by how much."

**Questions:**

1. In Fig. 1(h), BAxUS reaches about −430 on Humanoid at ~1k evaluations, while the BAxUS paper (Fig. 9) reports about −520 at a similar budget. Is this due to the hyperparameter settings?
2. In Fig. 1, could you report the wall-clock time or the best value at fixed wall-time budgets?
3. Line 212 states that starting GRABBO from 1D is more efficient, but the evidence shown is for BAxUS (1D vs 2D), where the gap is small on Fig. 2. Can you include GRABBO(2D-init) vs GRABBO(1D-init) on representative tasks?
4. Can you provide an error bound or sufficient conditions (kernel class, smoothness, noise) under which the surrogate regret used for the allocation tracks the true regret?
5. Does the $D^\star$ cap ever prevent reaching a subspace that adequately contains/approximates the optimizer? Additionally, could you report ablations with different $D^\star$ (e.g., 256, 4096, $\infty$)?

---

### Official Review · Reviewer_qsoL · 2025-11-01

**Soundness:** 2
**Presentation:** 2
**Contribution:** 2
**Rating:** 2
**Confidence:** 3

**Summary:**

This paper tackles the long-standing challenge of Bayesian optimization (BO) in high-dimensional spaces by revisiting the subspace assumption – the idea that the objective effectively lies on a low-dimensional manifold within the high-dimensional input space. The proposed algorithm, GRABBO, projects the search into a low-dimensional random subspace and dynamically expands that subspace over the course of optimization. Unlike prior subspace-based methods that often fix a small embedding or expand on a simple schedule, GRABBO uses a cumulative regret-guided criterion to decide when to increase the subspace dimension.

**Strengths:**

Clear algorithmic framing: The paper pinpoints BAxUS’s reliance on a failure-count threshold τ_fail as the source of early over-expansion in low-d and sluggish adaptation in high-d, motivating a budget-driven alternative. This is a precise, actionable critique that informs the design.

Clear scope of the experiment design: tested on eight problems and demonstrated the convergence with regret plots.

**Weaknesses:**

All chosen benchmark tasks exhibit some form of low effective dimensionality or structure – e.g., the Lasso problems have sparse true features by design, and classic functions like Hartmann6 or Branin have only a few important dimensions. The absence of a truly “dense” high-dimensional test case (where many or all dimensions contribute meaningfully to the outcome) means the evaluation may be optimistic. In settings without a clear low-dimensional manifold or sparsity (for instance, a function that depends on dozens of variables in a complex way), it’s unclear if GRABBO would still perform “great” or if it would struggle similarly to vanilla BO. The paper would be stronger if it acknowledged this and, if possible, tested a more adversarial high-D function without an obvious subspace structure.

The novelty of GRABBO, while well-executed, needs to be better established. The core idea oc progressively expanding a random subspace during BO is not entirely new. Prior methods have already explored dynamic subspace dimensionality: for example, BAxUS introduced nested subspaces that grow over time, and BOIDS similarly combines subspace embeddings with incremental search along structured directions. GRABBO’s contribution largely refines these ideas (identifying that BAxUS’s expansion trigger was too aggressive) and tweaks the strategy by using a regret-based expansion schedule. While this is a sensible improvement, it can be seen as an incremental refinement rather than a fundamentally new paradigm. The use of a trust region with success/failure counters is directly adopted from prior work (TuRBO/BAxUS), and the notion of allocating more budget to higher-dimensional subspaces aligns with intuitive extensions of BAxUS (which was budget-aware but used a simpler heuristic). Thus, some reviewers may feel the paper is an evolutionary step rather than a revolutionary one. It would strengthen the work if the authors clarified in detail how their approach substantially diverges from BAxUS (2022) and other prior work beyond the regret criterion. As it stands, the method blends known techniques and the magnitude of novelty needs to be better established.

Although the experiments include many baselines, there are notable high-dimensional BO approaches missing from the comparison, which raises concerns about the completeness of the evaluation. In particular, the paper does not compare against methods based on sparsity and variable selection in GPs, such as the SAASBO method

Incomplete experimental specification for reproducibility: Random seed policy is not stated in the main text (same/different seeds across methods; fixed vs. varied). No direct evidence was found in the manuscript. The acquisition function is only briefly mentioned as Thompson sampling; there is an absence of an ablation over UCB/EI/TS or justification for the choice limits interpretability (Sec. 5 mentions TS for BAxUS; GRABBO’s own default is not exhaustively justified).

Limited statistical analysis: Results are mean ± standard error over 10 runs, but there is no mention of significance testing (e.g., Wilcoxon, t-tests) or rank aggregation across tasks.

The comparative analysis against BAxUS and SBO baselines lacks sufficient diagnostic depth. Although the paper acknowledges BAxUS's limitations, it omits controlled experiments that would isolate the impact of budgeted versus failure-count triggers under consistent conditions (e.g., identical TR rules, acquisition methods, and seeds). The manuscript also lacks explicit runtime/scalability profiling across methods (e.g., wall-clock plots), a common practice highlighted in papers like “Standard GP…”, and no direct evidence to support these claims was found.

**Questions:**

Wrong citation: This seems to be a fictitious paper: “Leonard Papenmeier, Erik Daxberger, and Carl Edward Rasmussen. Baxus: Budget-aware sub-space search for high-dimensional bayesian optimization. In International Conference on Machine Learning, pp. 17112–17126. PMLR, 2022a,” as I could not find it on OpenReview or Google Scholar for paper or authors. The correct BAxUS paper is “Increasing the scope as you learn: Adaptive bayesian optimization in nested subspaces” which you included. If an LLM was used for finding a reference, this should be disclosed. Please also clean the references. Some have a URL and some do not. Format inconsistency is a problem.


Insufficient experiment: Only 10 independent runs. This is not enough. BAxUS and Standard GP, for example, repeat 20 times per run.


Insufficient problems benchmark against: In the original BAxUS paper, it also tested on problems such as Mopta08, SVM, and synthetic problems from Lassobench. In the SBO paper, they also have tested on the Rover problem. However, this paper is missing the study of these problems. Adding tests on these problems will make GRABBO more convincing.


The Synthetic Benchmark's Limitations: While the use of Branin, Hartmann, and Levy benchmarks with embedded effective dimensionalities is understood for testing low-dimensional subspace identification, it raises concerns about GRABBO's broader applicability. Does this imply GRABBO is only effective in such problems? This seems overly restrictive. It would be valuable to see GRABBO tested on problems with genuinely high dimensions, such as Levy with 400 dimensions or Ackley with 500 dimensions, to assess its effectiveness when all dimensions contribute to the optimization landscape.


Seed policy not stated: No record of the random seed fixing. Whether seeds are shared across methods per run should be listed; publish seed lists. Need statistical testing since random seeds have a significant impact on BO.


Add statistical rigor: A statistical ranking (e.g., Wilcoxon, Friedman) of algorithms will make the paper's case for the advantages of methods across problems. See how these papers perform ranking: https://arxiv.org/pdf/1809.04356, https://arxiv.org/pdf/2305.17535
Subtle differences in performance: In Figure 1, we see that GRABBO has very similar best-found optimal as other baseline algorithms in Branin2_500, LassoDNA, and HalfCheetah. It is not clear to me which is better due to the very close wins. It remains unclear what the margin of (dis)advantage is. Does it outperform other methods by a lot or is the performance difference negligible?


Runtime not recorded: In section B.4, you mentioned “the BAxUS method suffers from extreme runtime inefficiency in high dimensions.” However, there is no runtime record of how long each method takes to run the problem per iteration. Also, you are implementing more algorithm designs to GRABBO. How does this affect your runtime? If possible, please include runtime/scalability plots (e.g., function value vs. wall-clock time) to complement the iteration-based results, aligning with best practices.


Inconsistent iteration across methods: I understand there are computational limitations, but it is unfair to run baxus and boid with fewer iterations than your method, since they also appear to be still converging in Figure 1. Why not run it longer for a fair comparison?


Missing Acquisition ablations: Why Thompson sampling (TS) as the acquisition? There are also no details on implementing the TS sampling included in the paper. Please also formalize the acquisition optimization within TR. Moreover, have you done ablation studies of other acquisitions? (e.g., Expected Improvement (there is a BAxUS EI version on the BAxUS botorch website https://botorch.org/docs/tutorials/baxus/), UCB, ……)

**Details Of Ethics Concerns:**

This paper cited a paper that might not exist. Does this mean it should be desk-rejected? This is the strange citation that I can’t find anywhere: “Leonard Papenmeier, Erik Daxberger, and Carl Edward Rasmussen. Baxus: Budget-aware sub-space search for high-dimensional bayesian optimization. In International Conference on Machine Learning, pp. 17112–17126. PMLR, 2022a”)

---

### Official Review · Reviewer_voxi · 2025-11-01

**Soundness:** 3
**Presentation:** 3
**Contribution:** 2
**Rating:** 4
**Confidence:** 3

**Summary:**

The paper studies the use of random embeddings in high-dimensional Bayesian optimization. Past work (BAxUS) has allowed the dimensionality of the subspace to increase throughout the optimization. This paper takes that same approach, but develops new strategies for determining when to increase the subspace. Specifically, BAxUS uses a fail count mechanism that is often poorly calibrated and results in expansion of the search space too early. The new method, GRABBO, fixes this by replacing fail counts with an expansion schedule that is heuristically motivated by considerations of regret. Empirical evaluation shows that with this (and several other tweaks), GRABBO significantly outperforms BAxUS, and a number of other baseline methods.

**Strengths:**

* The detailed review of BAxUS in Section 3 was very helpful. The paper does an excellent job of elucidating the issues with BAxUS, and clearly identifying the difference between BAxUS and the new method, GRABBO.

* Empirical results show that these changes, while seemingly small, lead to a significant improvement in optimization performance.

* The paper is generally clear and well written.

* The ablation studies in the appendix are insightful.

**Weaknesses:**

* The paper claims that the expansion rate of BAxUS is set poorly, an that the new heuristic provides a better expansion rate. However, this is shown only anecdotally. The paper would definitely benefit from including results that show, e.g., the frequency of embedding expensions as a function of iteration, for both BAxUS and GRABBO.

* Ultimately the paper is building on BAxUS, and replacing one heuristic for embedding expansion with another. Now, the new heuristic is well-motivated from a regret formulation, but that is largely just motivation as a number of simplifying modifications are made and the paper is very clear that it is, by the time finished, just a heuristic. The improvement in optimization performance is great to see and speaks to the usefulness of the work. I do think that developing heuristics like this is important work that needs to be done, and can absolutely significantly improve the state of the art. But, it isn't the type of work that is usually published in ICLR.

**Questions:**

N/A

---

### Note · Authors · 2025-11-12

I have read and agree with the venue's withdrawal policy on behalf of myself and my co-authors.